# Mobile App-Based Health Promotion Programs: A Systematic Review of the Literature

**DOI:** 10.3390/ijerph15122838

**Published:** 2018-12-13

**Authors:** Mikyung Lee, Hyeonkyeong Lee, Youlim Kim, Junghee Kim, Mikyeong Cho, Jaeun Jang, Hyoeun Jang

**Affiliations:** 1College of Nursing, Yonsei University, 50-1, Yonsei-ro, Seodaemun-gu, Seoul 03722, Korea; lmk425@yuhs.ac (M.L.); goshimak@naver.com (Y.K.); zzomi324@naver.com (M.C.); je8952@naver.com (J.J.); scarlet311@hanmail.net (H.J.); 2Mo-im Kim Nursing Research Institute, College of Nursing, Yonsei University, 50-1, Yonsei-ro, Seodaemun-gu, Seoul 03722, Korea; kjh129@outlook.kr

**Keywords:** health promotion, mobile app, app-based intervention, smartphone

## Abstract

This study investigated the features and usefulness of mobile app-based health promotion programs for the general population. A comprehensive bibliographic search of studies on health promotion programs using mobile apps in peer-reviewed journals published in English up to November 2017 was performed using the PubMed, Embase, and CINAHL databases. The inclusion criteria were (1) randomized control trial designs; (2) assessed mobile app-based interventions to promote adult health conditions; 12 studies were ultimately included. The most common topics were diet and physical activity (*n* = 8) and overall healthy lifestyle improvement (*n* = 4). The purpose of the apps included providing feedback on one’s health status (*n* = 9) and monitoring individual health status or behavior change (*n* = 9). Across all studies, health outcomes were shown to be better for mobile app users compared to non-users. Mobile app-based health interventions may be an effective strategy for improving health promotion behaviors in the general population without diseases. This study suggests that mobile app use is becoming commonplace for a variety of health-promoting behaviors in addition to physical activity and weight control. Future research should address the feasibility and effectiveness of using mobile apps for health promotion in developing countries.

## 1. Introduction

Before the emergence of mobile phones, health care services were usually provided face-to-face with patients. Recently, however, medical and nursing interventions using mobile phones and apps have increased [1,2]. In particular, mobile app-based health promotion programs (hereinafter abbreviated as “mobile app programs”) are said to be an ideal platform for efficient interventions [3,4,5] because such mobile apps provide an easy way to access the target group [6] and are cost-effective compared to phone-based [7] and clinic-based [8] interventions. Currently, mobile phones are an important as well as a popular communication tool throughout modern society. In earlier studies, mobile apps were used for providing health education information [9,10], self-reporting [11], monitoring [10,12,13], data collection [14,15], and providing feedback [13,16] and notifications of visitation times [17]. Due to the advantage of real-time feedback, access through apps can facilitate participants in engaging in research, receiving individual education [18], generating sustained motivation through automatic sending of text messages and contact between users and health professionals [19,20], as well as ultimately changing health behaviors, as shown in a systematic review on physical activity [21].

Particularly, mobile app programs have been identified as useful tools for improving the efficiency of patients’ self-care as well as symptom management. A review of mobile app programs for diabetes showed that patients were able to measure their own blood glucose and easily check entered data with a smartphone, send data to medical staff in real time, and provide appropriate feedback [18]. Another review of a mobile app program for patients with alcoholism provided information on symptom improvement, motivational content, and tools for social support and drinking monitoring [22].

In recent years, the need for health promotion programs across the general healthy population has increased [3,23], and mobile app programs have been used to prevent and manage risk factors, increase physical activity, improve dietary habits [24], promote weight loss [25], and reduce smoking, stress, depression, and obesity [3]. In a study of cardiovascular screening, mobile app programs were more cost-effective than traditional methods such as paper-based or a mobile phone-based screening [8]. Often, individuals report having difficulty accessing health promotion programs, including advice, information, feedback, and self-monitoring, given the fast pace of modern life; hence, mobile app programs could provide an alternative [25,26]. For example, in a recent literature review, it was found that mobile app programs were effective in improving physical activity and healthy eating habits [27]. However, as this aforementioned review was limited to interventions to improve physical activity and diet in children and adults, it could not adequately determine the benefits of mobile app use for comprehensive health promotion behavior within adults. Adult population is an important group in terms of prevalence of health risk behaviors and the potential for scalability and wide dissemination of mobile app programs. Thus, rather than focusing on the efficacy of mobile app targeting a specific health behavior, it is important to know how and to what extent mobile apps have been used, their purposes, and how these apps have influenced various health behaviors in adults.

As a preliminary step prior to the development of mobile app health promotion programs, the objective of this study was to investigate the content and usefulness of mobile app programs for the general adult population.

## 2. Materials and Methods

### 2.1. Literature Search

A literature search was performed using PubMed, Embase, and CINAHL databases. Combinations of the following MeSH terms (“mobile applications”, “health promotion”) and keywords (“smartphone applications”, “app-based intervention”) were used to search for appropriate studies: [mobile applications AND health promotion] OR [smartphone applications AND health promotion] OR [app-based intervention AND health promotion]. The language was limited to English, and the search period was all-inclusive up to November 2017. No restrictions were placed on any specific health promotion behaviors, that is, walking, weight control, physical activity, sun protection behavior, etc. because this review intended to investigate to what extent mobile apps have been used in health promotion interventions for the general population. A total of 191 randomized controlled trial (RCT) studies were found through an initial search.

### 2.2. Study Selection

Studies were selected for review based on the following inclusion criteria: (1) use of RCT designs, (2) peer-reviewed journals, (3) published in English, and (4) concerned mobile app-based interventions to promote adult health. The following exclusion criteria were used: (1) non-RCT studies, (2) studies that used mobile phones for sending Short Message Service (SMS) messages or web-based interventions, (3) studies that did not aim at health promotion of the general population such as screening health status, testing mobile app validity, playing games, or using Twitter, and (4) studies targeted to patients with a specific disease as these may not be applicable to the general population. Two independent investigators (Mikyung Lee and Jaeun Jang) judged article eligibility and resolved discrepancies through a third reviewer (Hyeonkyeong Lee) and consensus-based discussion.

### 2.3. Data Extraction and Analysis

Two independent reviewers (Mikyung Lee and Jaeun Jang) used a custom form developed to extract appropriate data. The following categories were extracted: major field of research, sample size, setting, authors, publication year, countries, number of participants, app name and platform, purpose of the app, intervention period, and outcomes. According to an app’s purpose, articles were categorized as providing information, feedback, monitoring behavior change, and alarming the user regarding health behavior. Disagreements were resolved by a third reviewer (Hyeonkyeong Lee) and consensus-based discussion.

### 2.4. Study Quality Assessment

Two authors (Mikyung Lee and Mikyeong Cho) independently assessed the methodological quality of the included RCTs using Cochrane’s risk of bias tool [28]. Any disagreements with quality ratings were settled by consensus-based discussion. The analysis was performed using Review Manager 5.3 software (the Cochrane Collaboration, 2014, the Nordic Cochrane Centre, Copenhagen, Denmark).

The risk of bias tool includes the domains of selection bias (random sequence generation, allocation concealment), performance bias (blinding of participants and personnel), detection bias (blinding of outcome assessment), attrition bias (incomplete outcome data), and reporting bias (selective reporting). Each risk of bias was classified as low risk, unclear risk, and high risk of bias.

## 3. Results

### 3.1. Study Selection

A flow diagram for the study selection process is shown in Figure 1. The database searches produced 153 records after removal of duplicates. One hundred twenty items were excluded (e.g., review articles, qualitative studies, patients with a specific disease, using only SMS or web-based intervention (*n* = 85), no health promotion programs (*n* = 35)). After reviewing the abstracts, 33 full-text versions of studies were selected for further assessment of eligibility. Twenty-one of them were excluded because they failed to meet the inclusion criteria: the assessment of mobile-app validity (*n* = 3); mixed method, quasi-experimental design, RCT plan (*n* = 6); patient group (*n* = 6); and web-based, SNS intervention (*n* = 6). This led to the inclusion of 12 studies.

### 3.2. Result of Quality Assessment

The risk of bias assessment showed that all trials were at low risk of bias in most domains (Figure 2). All 12 papers reviewed through quality assessment were identified as low risk, and no articles were excluded. All 12 studies were assessed as having a low risk of selection bias (random sequence generation, allocation concealment). Two trials were considered to have a high risk of bias in the domain of double blinding, and one trial had an unclear risk of bias. Detection bias (blinding of outcome assessment) was assessed for two trials with high risk of bias, one trial as unclear risk, and the remainder as low risk. Attrition bias (incomplete outcome data) was determined by a dropout rate of 20%, which was found in two studies, and one study had an unclear risk. Reporting bias (selective reporting) was considered to be at high risk in two studies and unclear in three studies.

### 3.3. General Study Characteristics

The 12 RCT studies selected were all published in journals over the preceding five years (2013–2017). All studies were conducted in high-income countries. Four studies were in the U.S., and one each was in Denmark, England, Ireland, Canada, Australia, South Korea, Israel, and Singapore, respectively. In eight studies (66.7%), the major field was medicine, nursing, or public health; in four studies (33.3%), the major field was nutrition or exercise and sports. Four studies only targeted young adults under the age of 35. Five studies (41.8%) had a sample size of less than 100, three studies (25.0%) had a sample size of 100 to 200, two studies (16.6%) had 201–300, and two studies (16.6%) had a sample size over 300. All studies were conducted in community settings (Table 1).

For comparison, the design of seven studies used two groups (an experimental group applying a mobile app and a control group), and five studies were designed to use three groups or more. The minimum and maximum intervention period was 4 weeks and 24 months, respectively. Ten studies (83.3%) had an intervention period of less than 6 months (Table 2).

### 3.4. Usefulness of Mobile App-Based Health Promotion Interventions

Mobile apps are used for various purposes, including monitoring the health status, providing feedback, and providing health information. Apps were used to monitor individual health status or behavior change (75.0%; 9 of 12), provide feedback on one’s health status (75.0%; 9 of 12), or provide health-related information (66.7%; 8 of 12). Most of the mobile apps had been developed and used by the study team based on the study’s purpose, but some used previously developed app programs. No two studies used the same app. The primary study outcomes were significantly improved after the mobile app intervention. Intervention groups were provided a mobile app and additional interventions such as SMS, phone calls, group education, and a pedometer (66.7%; 8 of 12). In 8 out of 12 studies (66.7%), control groups were not provided an intervention while the remaining studies (33.3%) had control groups provided with a mobile app or website, diary, and pedometer.

#### 3.4.1. Diet and Physical Activity

In total, 8 studies implemented and described app interventions intended to improve healthy diet and physical activity. The basic functions of the apps included providing personalized feedback, practical tips, and tricks based on participants’ self-reported diet and exercise status [12,29]. Balk-Moller et al. [12] utilized an application (SoSu-life) developed to help reduce body weight in specific groups of employees within workplace settings. The SoSu-life app was further used to help participants reach their personal goals and provide “colleague challenges” and “weekly challenges” to interact and compete with other groups.

Carter et al. [30] developed an app for dietary control through an evidence-based behavioral approach and utilized the data for overweight adults; the app was applied for goal setting, dietary and physical activity self-monitoring, and feedback via weekly text messages. Another intervention study assessing weight loss for obese adults used an app to communicate interventions through goal setting, challenge games, and social support via a buddy system, which enabled self-monitoring from participants [31].

Electronic diaries were provided through another program to facilitate automatic self-monitoring of weight, activity, and caloric intake to prevent diabetes by reducing weight among overweight adults [10]. Additionally, the app used in this study provided reminders for participants to enter information every day and to deliver interactive intervention content, including daily messages, video clips, and quizzes. A study designed to improve eating behaviors used a food record app (mFR) to enable participants to easily record dietary intake data by taking pictures of food and drinks with a camera both before and after eating [17].

In an intervention study using an app to enhance physical activity, automatic feedback and tracking of step counts and calories burned was provided [14]. Another study [32], which used apps in an interesting way to promote physical activity, applied three different apps based on motivational frames drawn from behavioral science theory and evidence. The first one, called the “Analytic app”, focused on personal and quantitative goal setting, behavioral feedback, and provision of information to facilitate behavioral change. The user’s goal achievement was revealed through colorful meters. The second app, the “Social app” developed from a social influence perspective to strengthen social support for behavioral change, provided social normative feedback and emphasized modeling of behaviors by similar others and group-based collaboration and competition. The third app, called the “Affect app”, used principles, such as reinforcement scheduling and a bird avatar, to track the user’s physical activity.

#### 3.4.2. Other Health-Promoting Behaviors

In total, 4 studies implemented and described app interventions intended to produce changes in health-related behaviors (sun protection, vitamin D intake, bone health promotion, and coronary heart disease prevention). Buller et al. [13] developed an app (Solar Cell) for sun protection. The app provided current sunlight danger levels (low, medium, and extremely high), so as to warn users of sunburn damage, expected time for reapplying sunscreen, amount of vitamin D generated in the skin, and other advice for sun protection (i.e., wearing sunglasses, applying sunscreen, etc.). Furthermore, the app notified users of real-time UV levels depending on the user’s location and time.

Another app was designed to improve vitamin D intake among young adults. This app provided immediate feedback using a pie chart so that participants could compare their vitamin D and calcium intake levels to recommended levels, after entering food, drink, and supplements containing vitamin D and calcium that had been consumed. Time spent in the sunshine was also accounted for [16]. Additionally, helpful information was provided by automatically linking participants to the daily UV forecast once a postal code was entered [16].

A study using an app to improve bone health in young adult women with low bone mass provided achievement scores and feedback based on participants recording exercise, nutrition, and activity hours [15]. A study to prevent coronary heart disease provided four learning modules (physiology of the heart, prevalence of CHD/cardiac risk factors, information on healthy-heart lifestyles, and stress management) that allowed workers to learn within 20 min and obtain information based on three calculations (body mass index, daily caloric-intake, and 10-year CHD risk) [9].

### 3.5. Mobile App-Based Health Promotion Intervention Outcomes

Health outcomes included increased physical activity, promoting weight control, and reducing chronic disease risk. One study employed a mobile app to help office workers achieve personal weight loss goals through individual benchmarks and team competitions for weight loss. Results revealed that after 38 weeks, the app users demonstrated significant decreases in body weight, body fat percentage, and waist circumference compared to a control group [12].

Participants who used an app to self-monitor their daily caloric intake based on specific weight loss goals showed significantly higher adherence to the intervention compared to a group who used websites or diaries; the app-user group also had more significant weight loss and body fat reduction [30].

One intervention group from another study showed a statistically significant decrease in weight, and a significant increase in physical activity, when compared to a control group who used only a pedometer. Moreover, hip circumference, blood pressure, and saturated fat and sugar-sweetened beverage intake declined [10]. It has been reported that dietary assessments and tailored feedback using a mobile food record reduces intake of energy-dense and nutrient-poor foods, as well as sugar-sweetened beverages, which facilitate weight loss [17].

A study [32] that assessed physical activity promotion among adults by comparing three different apps (analytic, social, and affect apps) revealed a significant increase in moderate-to-vigorous-intensity physical activity (MVPA) among users of the social app compared to users of the other apps.

An intervention study designed to promote healthy lifestyles observed an increase in physical activity hours, significant decreases in weight, improvements in diet quality scores, and increased knowledge of nutrition within participants using a web-based app compared to a control group [29]. Another study reported increases in knowledge of coronary heart disease and management behaviors for blood cholesterol within the intervention group [9].

Among participants who were asked to use a mobile app that provided information and advice on sunburn danger, 77% started the app and received feedback more than once. These individuals reported higher sun-protection practices (e.g., using wide-brimmed hats) 7 weeks after the intervention compared to a control group [13]. Another study [15] provided feedback through the app about one’s osteogenic index by calculating and achievement scores for exercise, nutrition intake, and healthy living habits. Two experimental groups showed more achievement regarding knowledge of the benefits of exercise and calcium compared with the control group. The two experimental groups also demonstrated higher results in the serum levels of calcium, vitamin D, and sclerostin compared to those of the control group.

In another study, participants were asked to record their eating habits using an app, and prompt feedback was provided to improve vitamin D intake. This intervention group demonstrated a significant increase not only in average vitamin D intake but also vitamin D knowledge and perceived importance of vitamin D when compared to a control group [16].

## 4. Discussion

The present study reviewed the purposes and characteristics of mobile app programs designed for general population use in community settings. The main topics of health promotion were related to increased physical activity, weight control, sunlight exposure control, vitamin D intake, and bone strengthening. The most common purpose of a mobile app was to monitor individual health status or behavior change, provide feedback on one’s health status, and provide health-related information. In earlier systematic reviews designed to assess patients with specific diseases, the purpose of the intervention focused on information provision, health monitoring, and symptom information and management among patients with bipolar disorder [33], monitoring patient’s blood glucose level for individuals suffering from diabetes [34], symptom information and real time feedback for patients with cancer [35], and symptom management for adolescents with mental health issues [36]. The mobile apps were used to help participants monitor and change their behaviors by using information and criteria provided by the app. With a focus on general population use, the present review observed similar goals as those used for patient samples.

Based on the present review, mobile app programs for the general population have mainly been used for weight control and physical activity. However, the utility of mobile apps can be applicable to other health behaviors, including contraceptive use, oral health, human papillomavirus (HPV) vaccination, and smoking cessation [37]. Additionally, in the context of stress and depression, mobile apps could help individuals use checklists to assess their symptom profiles and obtain necessary information for mental health treatment. Preventive health management is also expected to have a large impact, even among those who have not visited a clinic. Here, healthcare providers can easily manage target populations by releasing health information, which could more readily facilitate behavioral interventions.

In each study of this review, the most common forms of the mobile app programs was to provide health information and feedback (50%, 6 of the 12 studies) followed by only providing feedback (25%, 3 of the 12 studies) or information (16.6%, 2 of the 12 studies). Consistent with this, in another review [38], providing information in a physical activity app, which provided automatic tracking of steps taken and calories burned, graphs, tables, and statistics on physical activity performed, a BMI calculator, etc., showed more desirable health outcomes for participants. Along with “feedback”, helpful advice/tips for increasing physical activity appeared to result in a significant improvement in the targeted outcome of health promotion behavior. Moreover, other useful elements of mobile app programs included in the reviewed studies were setting alarms to prompt health behavior [10], social interaction including within-team collaboration, and between-team competition [12,31,32].

Enhancing a mobile app’s entertainment element could also assist with intervention efficacy. Rather than merely providing information, an app can introduce games or quizzes to help people remember their health information and improve their learning [39,40]. Based on our findings, future research should include strategies designed to accelerate behavioral changes through mobile apps for competition [12,32,41], visualization [17], social support [12,32], and entertainment (e.g., challenge games, quizzes, etc.) [6,31,32,39,40]. For example, a mobile app-based intervention was a successful way to have users compete with each other and provide social support for facilitating goal achievement [42], often referred to as “provide opportunities for social comparison” as a behavior change technique (BCT) [43]. Mobile apps can also foster positive motivation by visually expressing a user’s present state with an appealing image [44], referred to as “use of imagery” as a BCT domain [43]. Furthermore, provision of social support from family and friends through mobile apps can increase intervention efficacy [45]. Additional studies have shown that interventions in which participants monitor their own behavior and assess goal achievement without feedback, peer-to-peer influence, and social support can also be effective [20,44]. This suggests that monitoring one’s own health behavior and goal attainment is just as important as social contact for individual health behavior.

A few study limitations should be noted. First, caution is needed when applying these results to a specific population (e.g., older adults and individuals with disabilities) in terms of smartphone utilization rates. Except for the study by Goodman [16], the studies in this review did not consider age differences in intervention efficacy. Second, all of the reviewed articles were conducted in developed countries, and developing countries are experiencing steady increases in smartphone use, as well as chronic disease prevalence. Further research is needed to assess the utility of mobile apps for health interventions within the general populations of developing countries. Finally, the usefulness of the app programs was evaluated largely based on comparing results between intervention and control groups. Thus, it is difficult to distinguish between the benefits of the app programs and the benefits of the associated interventions. Therefore, it is necessary to compare app programs within the same study to better isolate the role of app use.

## 5. Conclusions

This study showed that mobile app-based interventions could be useful for improving various health promotion behaviors including diet and physical activity for the general healthy population. Most of the app interventions reviewed focused on monitoring health status and behavior change, as well as provide feedback or health-related information. App programs may be more effective when social support is advocated and when entertainment functions or visualization features are included. Future research should address mobile app efficacy for health promotion in developing countries.

## Figures and Tables

**Figure 1 ijerph-15-02838-f001:**
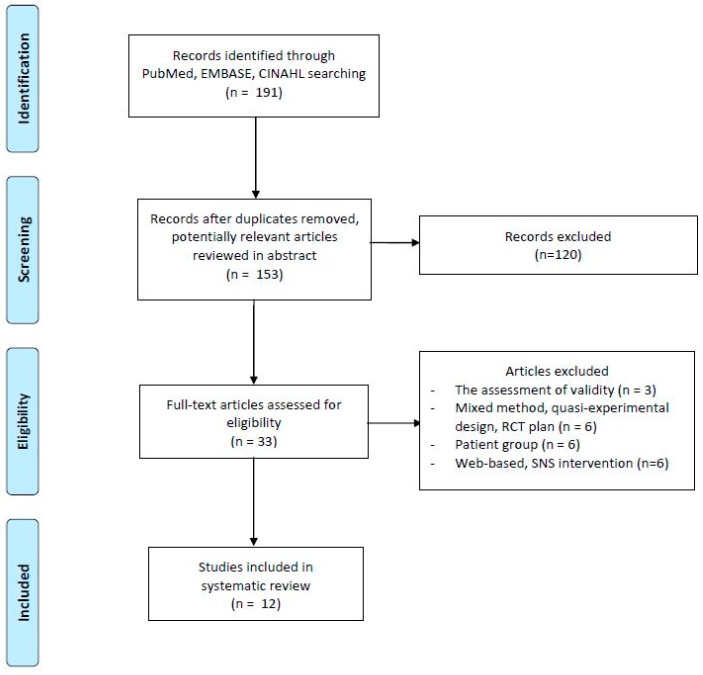
Study selection process.

**Figure 2 ijerph-15-02838-f002:**
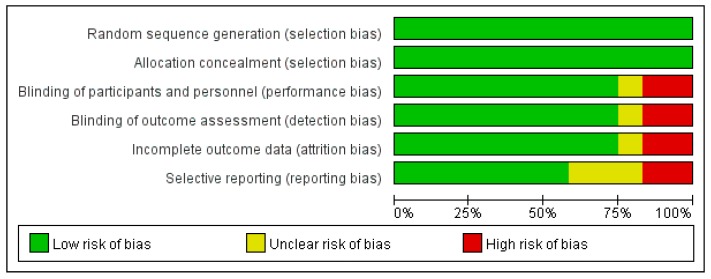
Cochrane’s risk of bias summary for health promotion apps reviewed.

**Table 1 ijerph-15-02838-t001:** General study characteristics (*n* = 12).

Variables	Categories	*n* (%)
Type of Studies	Published journal	12 (100.0)
Major field of researcher	Medicine (General, Nephrology, Dermatology)	4 (33.3)
	Nursing	3 (25.0)
	Nutrition, Exercise and Sports	4 (33.3)
	Public Health	1 (8.4)
Sample size	Under 100	5 (41.8)
	100–200	3 (25.0)
	201–300	2 (16.6)
	Above 300	2 (16.6)
Setting	Community	12 (100.0)

**Table 2 ijerph-15-02838-t002:** Features and outcomes of the app-based health promotion interventions (*n* = 12).

No.	Author, Year [Reference]	Sample Size			App Name	Platform	App Purpose	Intervention Period (Week)	Major Outcome Indices
		Total	Exp.	Cont.					
1	Balk-Moller et al., 2017 [12]	566	355 (App, E-mail, SMS, Communications in users)	211 (None)	SoSu-life	-	- Provide information, feedback - Monitor health status	16	- Body weight, body fat, waist circumference, blood pressure, total cholesterol
2	Buller et al., 2015 [13]	202	96 (App)	106 (None)	Solar Cell	Android, iOS	- Provide information, feedback - Monitor behavior change	8	- Sun protection practices, time spent outdoors, sunburn prevalence
3	Carter et al., 2013 [30]	128	43 (App, SMS, Photo)	42 (Website), 43 (Paper diary)	My Meal Mate (MMM)	Android	- Provide feedback - Monitor behavior change - Set goal	24	- Body weight, BMI, body fat
4	Fukuoka et al., 2015 [10]	61	30 (App, Pedometer)	31 (Pedometer)	Mobile Phone–Based Diabetes Prevention Program (mDPP)	iOS	- Provide information - Monitor health status - Alarm on the health behavior	20	- Body weight, BMI, hip circumference, blood pressure, lipid profile, glucose levels, daily steps, minutes per day
5	Glynn et al., 2014 [14]	90	45 (App, Call weekly)	45 (Call weekly)	Accupedo-Pro Pedometer app	Android	- Provide information, feedback - Monitor behavior change	8	- Daily step count, blood pressure, resting heart rate, body weight, mental health, qualityof life
6	Goodman et al., 2016 [16]	109	59 (App)	50 (None)	Vitamin D Calculator app (VDC-app)	iOS	- Provide information, feedback	12	- Intake, knowledge, perceptions of vitamin D, blood concentrations of 25(OH) D3
7	Kerr et al., 2016 [17]	247	82 (Dietary feedback and weekly SMS), 83 (Dietary feedback only)	82 (None)	Mobile food record (mFR) app	iOS	- Provide information, feedback - Monitor behavior change	24	- Intake of fruits, vegetables, energy-dense nutrient-poor foods and sugar-sweetened beverages, body weight, BMI
8	King et al., 2016 [32]	95	22 (Social app), 24 (Affect app), 22 (Analytic app)	27 (Control tracking diet app)	Analytically framed app, a socially framed app, an affectively framed app, or a diet-tracker control app	Android	- Monitor behavior change	8	- Duration of physical activity, sitting time
9	Park et al., 2017 [15]	103	36 (Mobile type bone health intervention),38 (Group education only)	29 (None)	Strong bone, Fit body (SbFb)	Android	- Provide feedback - Record	20	- Bone mineral density, minerals, biochemical, markers, food intake diary, knowledge, health belief, self-efficacy
10	Naimark et al., 2015 [29]	99	69 (App)	30 (None)	eBalance	web-based	- Provide information, feedback - Monitor behavior change	14	- Nutrition knowledge, diet quality, physical activity, weight, waist circumference
11	Svetkey et al., 2015 [31]	365	122 (Cell phone), 120 (Personal coaching)	123 (None)	CITY	Android	- Provide feedback - Monitor health status, behavior change	24months	- Body weight
12	Zhang et al., 2017 [9]	80	40 (App, SMS)	40 (None)	Care4 Heart	Android, iOS	- Provide information	4	- Knowledge of coronary heart disease, perceived stress level, cardiac-related lifestyle behaviors

Note: Exp. = experimental group, Cont. = control group.

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
