# Peer review of "Mobile App-Based Health Promotion Programs: A Systematic Review of the Literature"

_ijerph, 2018, doi:10.3390/ijerph15122838_

Round 1
Reviewer 1 Report
This is an interesting and well-written article that should be considered for publication. I have two recommendations for revisions that would strengthen the article, and both should be readily achievable.
First, I don't see any value in Figure 1, since this information is sufficiently described in the body of the manuscript. I suggest it be deleted.
Second, the authors should consider including at least a cursory assessment or interpretation about what behavior change/intervention elements appear to be important to effectiveness of the apps reviewed. For example, 8 of the 12 studies summarized addressed diet and exercise changes. Each app was unique in the intervention elements included (e.g., goal setting, self-monitoring of weight and activity, feedback, social reinforcement), however some common elements were shared by multiple apps. I would like to see at least a cursory examination of the intervention elements that appear to be important to a significant change in the targeted outcome measure. This summary could be included in the discussion following the next to last paragraph.
Thanks for the opportunity to review this manuscript
Author Response
This is an interesting and well-written article that should be considered for publication. I have two recommendations for revisions that would strengthen the article, and both should be readily achievable.
We thank your thoughtful comments. The responses to your comments have been written in red below.
First, I don't see any value in Figure 1, since this information is sufficiently described in the body of the manuscript. I suggest it be deleted.
We think that visualization of the risk of bias in included studies would be informative for readers to interpret the contribution of the information as showing the proportion of information that comes from studies at low, unclear, or high risk of bias for each item.
Second, the authors should consider including at least a cursory assessment or interpretation about what behavior change/intervention elements appear to be important to effectiveness of the apps reviewed. For example, 8 of the 12 studies summarized addressed diet and exercise changes. Each app was unique in the intervention elements included (e.g., goal setting, self-monitoring of weight and activity, feedback, social reinforcement), however some common elements were shared by multiple apps. I would like to see at least a cursory examination of the intervention elements that appear to be important to a significant change in the targeted outcome measure. This summary could be included in the discussion following the next to last paragraph.
In each study of this review, the most common element of the mobile app programs was to provide health information and feedback (50%, 6 of the studies) followed by only providing feedback (25%, 3 of the 12 studies) or information (16.6%, 2 of the 12 studies). Consistent with this, in another review [37], providing information in a physical activity app, which provided automatic tracking of steps taken and calories burned, graphs, tables, and statistics on physical activity performed, a BMI calculator, etc., showed more desirable health outcomes for adolescents. As well as “feedback,” helpful advice/tips for increasing physical activity appeared to be important to a significant change in the targeted outcome health promotion behavior. Moreover, other useful elements of mobile app programs included in the reviewed studies were setting alarms to prompt health behavior [10], social interaction including within-team collaboration, and between-team competition [12,30,32].
Reviewer 2 Report
It is a good review paper, which provides an overview of mobile healthcare apps.
As a review paper, I think this paper has presented well about the review.
Even selected literature is limited, the review of the selected papers has been done properly.
The paper provides good information about the "Materials and Methods", which shows the literature was carefully selected and demonstrates how a systematical review should be done.
The review results are well presented and discussed.
The conclusion based on the results make sense.
The author cloud re-design the table layout to have better presentation and to meet the format requirement of this journal.
Author Response
It is a good review paper, which provides an overview of mobile healthcare apps.
As a review paper, I think this paper has presented well about the review.
Even selected literature is limited, the review of the selected papers has been done properly.
The paper provides good information about the "Materials and Methods", which shows the literature was carefully selected and demonstrates how a systematical review should be done.
The review results are well presented and discussed.
The conclusion based on the results make sense.
The author cloud re-design the table layout to have better presentation and to meet the format requirement of this journal.
We thank your thoughtful review and comments. The manuscript has been changed in accordance with your comments regarding the table layout and journal format. Professional English editing service was also done.
Reviewer 3 Report
Dear editor and authors,
Thank you very much for the opportunity to review this systematic review. It provides some overview regarding the content of mobile phone apps as a health promotion strategy.
Please find the remarks below:
Abstract
The abstract looks very disorganized to me. Exact search-term should be provided in a separated section out of the abstract. Study flow is not presented fully. No effect-sizes are given. What does utility mean in this context? What exactly has future research to address?
Needs serious language and content editing.
Line 23+24: The authors wrote: “Health outcomes of all studies were effective than control groups“. This sentence should pinpoint the direction and strength of the relationship (e.g. “Across all studies, health outcomes have been shown to be more effective for mobile app users compared to the control group”). Also, the word “effective” does not seem well suited in this context and it remains unclear what constitutes the “control group”.
Introduction
The whole introduction section is very inconsistent and not well organized. Many terms referring to different constructs are mixed together (e.g. health care services, treatment options, and promotion).
Line 33: The reference given provides only limited evidence.
Line 35: The cost-effectiveness is NOT supported by the data presented in the reference.
Line 40: Lacking a reference.
Line 41-46: The structure of a paragraph is very messy. Check this reference for more information: Dunn, D. (2011). A short guide to writing about psychology. Harlow: Longman.
Line 47-58: The structure of this paragraph also needs some editing to be logically aligned. What do you want to express in this paragraph? It seems that you listed many relevant, but unrelated sentences after another. Try to structure your central statements, bring them in order and give them more context. This paragraph should guide the reader towards a clear presentation of the supporting rationale: What has been done in this review? Why? How can results from this review extend what we already know? Also, the abundant use of the word “and” should be avoided and replaced with “as well as”.
Line 59: Who is the local community? What do you exactly mean by “characteristics underlying the utility”? The overall purpose of the review is not clear. Are you interested in the content, quality, features or effectiveness in promoting healthy behaviors regarding mobile-based apps? Clearly, state the aim of the present paper and link it to your discussion (e.g. “The present paper was set out to investigate the content of app programs for the general population […]).
Materials and Method
Line 65: PRISMA is a guideline for reporting NOT for conducting reviews. Please read carefully when citing guidelines.
Line 65: Which MeSH terms did you use? The searchterm is not elaborated enough. Please check out the Cochrane review handbook to learn more about searchterm construction.
Line 69: What do you mean by “health promotion behaviors”?
Line 71-79: It remains unclear how the authors came to their in- and exclusion criteria. Did this follow any rationale? Did you publish any protocol ahead? Or a registration at PROSPERO? Can´t find this so there is a huge potential bias.
Line 77: Correct a minor spelling mistake (i.e. “heath” to “health”).
Line 78: Please state the researchers' names who performed the judgment.
Line 78+79: Why did you exclude patients with a specific disease? There could be health promoting apps for specific diseases? If you exclude them, because they do not reflect the “general population”, explain why. How many studies did you exclude? You could also be more specific about how the investigators judged article eligibility.
Line 81+86: Again, please say who. Why is it sometimes two and sometimes three researchers?
Line 87 ff: If results are presented, please do so in the results section.
Line 89: Provide a corresponding reference for the Cochrane tool.
Line 104: The assessment of the methodological quality led to what (e.g. exclusion)?
Line 105: Figure 2: I don´t understand what means “exclusion with reasons”. Did you also exclude without reason? Be more specific in the row “eligibility”. The author wrote: “Articles excluded With reasons (the assessment of Validity, […])”. This seems rather vague. Adjust the capital letters for correct and consistent spelling (i.e. “Reasons”, “Validity”).
Line 129: The layout of table 2 appears a little disorganized. It needs some formatting and consistency.
Line 141: Probably it would be helpful to transform this section into a table? You merely provide the content of the individual apps and the overall purpose remains unclear. What can you conclude from here?
Conclusion
Line 277: Correct the grammar of this sentence (e.g. “Mobile app interventions appeared to be effective for improving health promotion”). On which empirical basis do you conclude that mobile app interventions appeared to be effective for improving health promotion?
Author Response
We thank your thorough review and constructive comments that .have enriched the manuscript and produced a better and more balanced account of the research. The manuscript has been changed in accordance with your comments as you seen in the table below. Professional English editing was also done.
Comments and Suggestions for Authors | Author's Notes to Reviewer | |
Abstract | ||
The abstract looks very disorganized to me. Study flow is not presented fully. No effect-sizes are given. Exact search-term should be provided in a separated section out of the abstract. What does utility mean in this context? | We added more information to the method and conclusion, and created a smoother flow by including connecting sentences. | |
(Lines 74-78) We deleted search-terms from the abstract. You will find them in the Methods section. | ||
Our interest was the features and usefulness of mobile app programs for the general population, not those with specific diseases. Therefore, we did not identify effect sizes. | ||
What exactly has future research to address? | We rephrased this sentence as follows: “This study suggests that mobile app use is becoming commonplace for a variety of health promoting behaviors in addition to physical activity and weight control. Future research should address mobile app efficacy for health promotion in developing countries.” | |
Needs serious language and content. editing | We have received professional editing. | |
Line 23+24: The authors wrote: „Health outcomes of all studies were effective than control groups.“ This sentence should pinpoint the direction and strength of the relationship (e.g. “Across all studies, health outcomes have been shown to be more effective for mobile app users compared to the control group”). | In accordance with the reviewer’s comment, we corrected the sentence as follows: (Lines 25) “Across all studies, health outcomes were shown to be better for mobile app users compared to non-users.” | |
Also, the word “effective” does not seem well suited in this context and it remains unclear what constitutes the “control group”. | (Line 25) We substituted the word “better” for “effective.” We substituted the word “non-users” for “control group.” | |
Introduction | ||
The whole introduction section is very inconsistent and not well organized. | We revised and reconstructed the entire introduction section. | |
Many terms referring to different constructs are mixed together (e.g. health care services, treatment options, and promotion). | (Lines 58-61) For consistency, we substituted the word “mobile app program” in place of the terms “app-based program” and “mobile-app programs.” (Line 161) Additionally, the word “web-based” was revised. | |
Line 33: The reference given provides only limited evidence. | We added two further references (4,5). | |
Line 35: The cost-effectiveness is NOT supported by the data presented in the reference. | In the literature cited, mobile app programs were found to be cost effective compared to phone-based and clinic based interventions. | |
Line 40: Lacking a reference. | We added the following references: Riley, W.T.; Rivera, D.E.; Atienza, A.A.; Nilsen, W.; Allison, S.M.; Mermelstein, R. Health behavior models in the age of mobile interventions: are our theories up to the task? Translational behavioral medicine 2011, 1, 53-71, doi:10.1007/s13142-011-0021-7. Ferguson, C.; Jackson, D. Selecting, appraising, recommending and using mobile applications (apps) in nursing. Journal of clinical nursing 2017, 26, 3253-3255, doi:10.1111/jocn.13834.. | |
Line 41-46: The structure of a paragraph is very messy. Check this reference for more information: Dunn, D. (2011). A short guide to writing about psychology. Harlow: Longman. | (Line 46-52)We have rearranged the sentences for greater clarity as follows: Especially, mobile app programs have been identified as useful tools for improving the efficiency of patients’ self-care as well as symptom management. A review of mobile app programs for diabetes showed that patients were able to measure their own blood glucose and easily check entered data with a smartphone, send data to medical staff in real time, and provide appropriate feedback [18]. Another review of mobile app program for patients with alcoholism provided information on symptom improvement, motivational content, and tools for social support and drinking monitoring [22]. | |
Line 47-58: The structure of this paragraph also needs some editing to be logically aligned. What do you want to express in this paragraph? It seems that you listed many relevant, but unrelated sentences after another. Try to structure your central statements, bring them in order and give them more context. This paragraph should guide the reader towards a clear presentation of the supporting rationale: What has been done in this review? Why? How can results from this review extend what we already know? | We have added further content to assist readers’ understanding. | |
Also, the abundant use of the word “and” should be avoided and replaced with “as well as”. | We replaced the word “and” with “as well as.” (Lines 39, 44, 47) | |
Line 59: Who is the local community? | (Line 69) We wanted to identify evidence of the usefulness of mobile app programs for healthy populations. So we changed the term “local community” to “general adult population.” | |
What do you exactly mean by “characteristics underlying the utility”? The overall purpose of the review is not clear. Are you interested in the content, quality, features or effectiveness in promoting healthy behaviors regarding mobile-based apps? Clearly, state the aim of the present paper and link it to your discussion (e.g. “The present paper was set out to investigate the content of app programs for the general population […]). | The purpose of this study was rephrased as follows: As a preliminary step prior to the development of mobile app health promotion programs, the objective of this study was to investigate the content and usefulness of mobile app programs for the general adult population. | |
Materials and Methods | ||
Line 65: PRISMA is a guideline for reporting NOT for conducting reviews. Please read carefully when citing guidelines. | We deleted the mention of PRISMA. | |
Line 65: Which MeSH terms did you use? The search term is not elaborated enough. Please check out the Cochrane review handbook to learn more about search term construction. | (Line 75-76)The MeSH terms we used were “mobile applications, health promotion” as well as the keywords “smartphone applications,” “app-based intervention.” | |
Line 69: What do you mean by “health promotion behaviors”? | We specified that “health promotion behavior” includes walking, weight control, physical activity, sun protection behavior, etc. | |
Line 71-79: It remains unclear how the authors came to their in- and exclusion criteria. Did this follow any rationale? Did you publish any protocol ahead? Or a registration at PROSPERO? Can´t find this so there is a huge potential bias. | The purpose of this study was to investigate the content and usefulness of mobile app programs that used in general population rather than focusing effectiveness of mobile app for specific target outcome. Through this review of the scientific literature, we wanted to show studies that demonstrate how such mobile app programs have been constructed, and the results that have been obtained, when using mobile apps. | |
Line 77: Correct a minor spelling mistake (i.e. “heath” to “health”). | (Line 89) We corrected the spelling. | |
Line 78: Please state the researchers' names who performed the judgment. | (Line 91-92) Two independent investigators (M.L. and J.J) judged article eligibility and resolved discrepancies through a third reviewer (H.L) or consensus-based discussion. | |
Line 78+79: Why did you exclude patients with a specific disease?: There could be health promoting apps for specific diseases? If you exclude them, because they do not reflect the “general population”, explain why. | There are several app-based interventions for patients with specific diseases. In this study, we wanted to investigate the contents and usefulness of mobile-app programs for the general population in community settings. | |
Line 79: (Contined) How many studies did you exclude? You could also be more specific about how the investigators judged article eligibility. | (Lin2 115-122)We described the study selection in the 1st paragraph of the Result section. After reading the full-text of the articles, 21 of 33 were excluded. | |
Line 81+86: Again, please say who. Why is it sometimes two and sometimes three researchers? | Two authors reviewed article eligibility and data extraction. Disagreements were resolved via a third reviewer and consensus–based discussion. For study quality assessment, two researchers reviewed the studies and disagreements were resolved through consensus-based discussion. | |
Line 87 ff: If results are presented, please do so in the results section. | (Line 123)We moved the subsection “Results of quality assessment” into the result section. | |
Line 89: Provide a corresponding reference for the Cochrane tool. | We added the following reference. 28. Higgins JPT, Altman DG, Gøtzsche PC, et al. The Cochrane Collaboration’s tool for assessing risk of bias in randomised trials. BMJ 2011;343:d5928. | |
Line 104: The assessment of the methodological quality led to what (e.g. exclusion)? | We stated that all 12 papers reviewed through quality assessment were identified as low risk, and no articles were excluded. | |
Line 105:Figure 2: I don´t understand what means “exclusion with reasons”. Did you also exclude without reason? Be more specific in the row “eligibility”. The author wrote: “Articles excluded With reasons (the assessment of Validity, […])”. This seems rather vagueAdjust the capital letters for correct and consistent spelling (i.e. “Reasons”, “Validity”). | We wrote the specific reasons for exclusions as follows:
Articles excluded - Validity assessment of mobile apps (n = 3) - Mixed method, quasi-experimental design, RCT plan (n = 6) - Patient group (n = 6) - Web-based, SNS intervention (n = 6) | |
We corrected the spelling. | ||
Results | ||
Line 129: The layout of table 2 appears a little disorganized. It needs some formatting and consistency. | The layout of the table was reorganized following the format of this journal. | |
Line 141: Probably it would be helpful to transform this section into a table? You merely provide the content of the individual apps and the overall purpose remains unclear. What can you conclude from here? | Two sections (“3.4” and “3.5”) were reorganized into the section “General Study Characteristics.” We wanted to describe the contents of mobile apps in more detail in terms of specific health promoting behaviors. | |
Conclusion | ||
Line 277: Correct the grammar of this sentence (e.g. “Mobile app interventions appeared to be effective for improving health promotion”). On which empirical basis do you conclude that mobile app interventions appeared to be effective for improving health promotion? | The sentence was rephrased to correct the grammar and state a more tentative conclusion as follows: “This study showed that mobile app based interventions could be effective for improving health promotion behaviors such as diet and physical activity.” | |
Round 2
Reviewer 3 Report
The authors did improve the article content and did language editing. The introduction section has improved, but is still little confusing to read. The search-term is not very elaborated, which limits the methodological quality of the study and influences the results. Within the results we see a huge variety of different approaches (e.g. health information, monitoring) and scopes (body weight, fitness, coronalry heart disease) mixed together. The overall contribution to the field seems very limited.
Author Response
Thank you for your constructive comments that have enriched the manuscript.
We tried to include the following in the Introduction:
-The importance and purpose of mobile-phone and apps in health research and practice
-Major roles of mobile app in patient groups
-The gap between the earlier reviews of patients or specific health behavior and current study.
We believe that the structure and logic of the Introduction is appropriate to address the significance of this study.
We used both MeSH terms (“mobile applications,” “health promotion”) and keywords (“smartphone applications,” “app-based intervention”) to search for appropriate studies without any restrictions for year of publication. We did not use MeSH terms or keywords for specific health behavior because this review intended to investigate to what extent the mobile app has been used in health promotion interventions for the general population.
We intended to investigate the versatility of the mobile app in health interventions. We found that it is used for several purposes, including providing health information and monitoring health status. The scope of outcomes also varied because we did not target any specific health behavior, and we did not limit our review to any specific health outcome or problem. We aimed to examine how the app was used and how it helped to improve the health-promoting or disease-prevention behaviors of the general healthy population.
Our study is a review of the content and usefulness of a mobile app to determine how and to what extent mobile apps have been used, their purposes, and how these apps have influenced various health behaviors in adults Our discussion suggests that other research teams will be able to develop mobile app applicable to the general population to encourage health-promoting behavior.